

# Retrospective study of transcriptomic profiling identifies Thai triple-negative breast cancer patients who may benefit from immune checkpoint and PARP inhibitors

Monthira Suntiparpluacha[1,*], Jantappapa Chanthercrob[1,*], Doonyapat Sa-nguanraksa[2], Juthamas Sitthikornpaiboon[1], Amphun Chaiboonchoe[1], Patipark Kueanjinda[3], Natini Jinawath[4,5,6] and Somponnat Sampattavanich[1]

[1] Siriraj Center of Research Excellence for Systems Pharmacology, Department of Pharmacology, Faculty of Medicine Siriraj Hospital, Mahidol University, Bangkok, Thailand
[2] Division of Head Neck and Breast Surgery, Department of Surgery, Faculty of Medicine Siriraj Hospital, Mahidol University, Bangkok, Thailand
[3] Center of Excellence in Immunology and Immune-mediated Diseases, Department of Microbiology, Faculty of Medicine, Chulalongkorn University, Bangkok, Thailand
[4] Program in Translational Medicine, Faculty of Medicine Ramathibodi Hospital, Mahidol University, Bangkok, Thailand
[5] Chakri Naruebodindra Medical Institute, Faculty of Medicine Ramathibodi Hospital, Mahidol University, Samut Prakan, Thailand
[6] Integrative Computational BioScience (ICBS) Center, Mahidol University, Nakhon Pathom, Thailand
* These authors contributed equally to this work.

Corresponding authors
Natini Jinawath,
natini.jin@mahidol.ac.th
Somponnat Sampattavanich,
somponnat.sam@mahidol.edu

## ABSTRACT

**Background**. Triple-negative breast cancer (TNBC) is a rare and aggressive breast cancer subtype. Unlike the estrogen receptor-positive subtype, whose recurrence risk can be predicted by gene expression-based signature, TNBC is more heterogeneous, with diverse drug sensitivity levels to standard regimens. This study explored the benefit of gene expression-based profiling for classifying the molecular subtypes of Thai TNBC patients.

**Methods**. The nCounter-based Breast 360 gene expression was used to classify Thai TNBC retrospective cohort subgroups. Their expression profiles were then compared against the previously established TNBC classification system. The differential characteristics of the tumor microenvironment and DNA damage repair signatures across subgroups were also explored.

**Results**. Thai TNBC cohort could be classified into four main subgroups, corresponding to the LAR, BL-2, and M subtypes based on Lehmann's TNBC classification. The PAM50 gene set classified most samples as basal-like subtypes except for Group 1. Group 1 exhibited similar enrichment of the metabolic and hormone response pathways to the LAR subtype. Group 2 shared pathway activation with the BL-2 subtype. Group 3 showed an increase in the EMT pathway, similar to the M subtype. Group 4 showed no correlation with Lehmann's TNBC. The tumor microenvironment (TME) analysis showed high TME cell abundance with increased expression of immune blockade genes in Group 2. Group 4 exhibited low TME cell abundance and reduced immune blockade

gene expressions. We also observed distinct signatures of the DNA double-strand break repair genes in Group 1.

**Conclusions**. Our study reported unique characteristics between the four TNBC subgroups and showed the potential use of immune checkpoint and PARP inhibitors in subsets of Thai TNBC patients. Our findings warrant further clinical investigation to validate TNBC's sensitivity to these regimens.

# INTRODUCTION

Breast cancer is a leading cause of cancer death in women worldwide. It is a very heterogeneous disease and is divided into three major types based on estrogen receptor (ER), progesterone receptor (PR), and human epidermal growth factor receptor-2 (HER2) status. Targeted therapeutic agents are available for ER+ and HER2+ breast cancers. Specifically, for ER+ breast cancer, a gene expression-based algorithm can further classify patients into five molecular subtypes and predict the recurrence risk (*Parker et al., 2009*). Unfortunately, there have not been standardized regimens apart from chemotherapy for the ER-/PR-/HER2- or so-called triple-negative breast cancer (TNBC), the most aggressive and heterogeneous type. Understanding the molecular distinction among TNBC patients is vital to developing effective tailored regimens for individual TNBC patients.

The first publication to demonstrate further subtyping of TNBC patients using gene expression data was by *Lehmann et al. (2011)*. These subtypes were called the Vanderbilt TNBC subtypes, consisting of basal-like 1 and 2 (BL1, BL2), luminal androgen receptor (LAR), mesenchymal (M), mesenchymal stem-like (MSL), and immunomodulatory (IM). Since then, several research groups have further validated these subtypes and narrowed them down to four major types (BL1, BL2, LAR, M) (*Burstein et al., 2015*; *Lehmann et al., 2021*; *Wang et al., 2019*; *Yin et al., 2020*; *Yoo et al., 2022*). Other follow-up studies also reported several new subclasses for TNBC classification (*Burstein et al., 2015*; *Ensenyat-Mendez et al., 2021*). Some subtypes within TNBC patients were sensitive to targeted therapies *in vitro*, providing hope for some subgroups of TNBC patients (*Lehmann et al., 2011*; *Yin et al., 2020*).

In this study, we examined the subgroups within our Thai TNBC cohort and investigated their molecular characteristics using NanoString nCounter® technology-based gene expression profiling. By mapping the identified subgroups with the prior TNBC classifications, we validated and identified key activated pathways, characterized the immunophenotypes, and compared the DNA repair activities across these subgroups. These results lead to a better understanding of the molecular heterogeneity among Thai TNBC patients, contributing to developing more tailored regimens for different TNBC subgroups.

## MATERIALS & METHODS

### Patient population

Specimens used in this study were selected from archival frozen tumor tissues in RNA*later*™ nStabilization Solution (Thermo Fisher Scientific, Waltham, MA, USA) (Division of Molecular Genetics, Faculty of Medicine, Siriraj Hospital archival specimens) and our tissue biobank. A total of 28 samples (with tissue size larger than $3 \times 3 \times 3$ mm³, known clinical data, histological information, confirmatory immunohistochemistry and fluorescence *in situ* hybridization of their ER-negative/PR-negative/HER-negative status) were selected from the bank. The specimens were collected after the patient signed the approved participant information sheet and informed consent form. The study's protocol was approved by the Siriraj Institutional Review Board (COA number Si 329/2017) following the international guidelines for human research protection (the Declaration of Helsinki and the International Conference on Harmonization in Good Clinical Practice).

### RNA purification and gene expression analysis using NanoString nCounter® platform

The frozen tissues were thawed to room temperature 30 min before RNA purification. The thawed tissues were transferred to new 1.5-ml microcentrifuge tubes containing 1 ml of GENEzol™ Reagent (Geneaid, New Taipei, Taiwan) and were ground using plastic pastel until homogenized. The ground tissues were incubated at room temperature with GENEzol™ Reagent for 5 min. Then, 200 µl of chloroform was added into each tube, and the tubes were vigorously shaken for 15 s and let stand vertically at room temperature for 2 min. The tubes were centrifuged at $12,000 \times g$ for 15 min at 4 °C, and the top RNA-containing clear solution was transferred to new tubes. The same volume of 70% ethanol was mixed into the clear solution, and 700 µl of the mixed solution was added into a spin column. The RNA purification process was performed using PureLink™ RNA Mini Kit (Ambion, Thermo Fisher Scientific) following the manufacturer's instructions. Concentrations and purity of the extracted RNA were measured with NanoDrop™ One Microvolume UV-Vis Spectrophotometer (Thermo Fisher Scientific).

The extracted RNA was subjected to gene expression evaluation with NanoString nCounter® platform (NanoString Technologies, Seattle, WA, USA). The nCounter® Breast Cancer 360™ Panel was used in this study, including breast cancer subtypes signatures, immune response and tumor microenvironment signatures, and key breast cancer pathways such as DNA damage repair, cell cycle, hormone receptor signaling, and cancer metabolism. In short, 100 ng of RNA, at A260/280 more than 1.7, was hybridized with Reporter Codeset and Capture ProbeSet at 65 °C for 18 h. The hybridized RNA was then transferred into the cartridge in the nCounter® Prep Station, and the number of probed mRNA was quantified with the nCounter® FLEX Digital Analyzer.

### Data analysis

Subgroup and pathway analyses were done using R software (version 4.0.3) and the Rosalind program. The mRNA counts were normalized with the NanoStringNorm package (*Waggott et al., 2012*) for further analysis. For subpopulation analysis, the normalized gene expression

 

data was reduced in its dimensions and computed group centroids by principal component analysis (PCA) and unsupervised k-means clustering to determine the optimal clustering groups, respectively. To specify the proper number of clusters, an elbow method was utilized for assessing the total within the sum of square values at various $k$ values. The selected $k$ should be the point where the area under the curve sharply decreases before the total within the sum of squares stabilizes (the sharpest elbow). Moreover, we also analyzed the average silhouette width values for different values of $k$ to confirm the appropriate number of clusters. To understand the similarity between our TNBC groups and the previously defined TNBC, we also determined the relation between our subgroups and the TNBC data set from Lehmann's subtypes (*Lehmann et al., 2021*) using the subclass mapping (SubMap) algorithm following the method from *Hoshida et al. (2007)* with a significant value $p < 0.05$. We acquired gene expression data of TNBC tumors with clinical information from TCGA and subtype calling from Lehmann's publication. To combine the two genomic data sets from different transcriptomic technologies, we used the ComBat function (*Johnson, Li & Rabinovic, 2006*) in the R software to minimize the nonbiological batch effect among these data sets. For the breast cancer intrinsic subtyping identification, our gene expression profiles were predicted by the PAM50 classifier (*Parker et al., 2009*), molecular.subtyping function (genefu library version 2.28.0) on the R software. To establish reference profiles, we used published NanoString gene expression data from *Bustamante Eduardo et al. (2019)* with known molecular subtypes and data from our archival specimens from ER-positive and HER2-positive patients as the reference specimens for our intrinsic subtype analysis. For pathway analysis, we utilized the Rosalind Platform for nCounter Data Analysis with a cut-off threshold of 1.5 fold-change and the adjusted p-value less than 0.05 to examine the activation of different cancer hallmark gene sets following the MSigDB Hallmark Pathway database (*Liberzon et al., 2015*). For the statistical analysis, the one-sided Wilcoxon Rank Sum test and the Kruskal–Wallis test were used for mean comparisons of two and multiple groups, respectively. A significant level of $p$-value was less than 0.05.

### Immune infiltration analysis

Tumor immune estimation resource (TIMER) is the public database used for investigating the relationship between cancer and immune cell infiltration using gene expression data. To determine the impact of immune cell infiltration in TNBC subgroups, the TIMER2.0 (*Li et al., 2020*; *Sturm et al., 2019*) with immune cell estimation using the Microenvironment Cell Population (MCP) Counter algorithm (*Becht et al., 2016*) was used to estimate immune infiltration. Such a tool enables the arbitrary unit approximation of tumor-associated immune cells (CD8+ T cells, B cells, myeloid dendritic cells, neutrophils), cancer-associated fibroblasts, and endothelial cells. The results are presented as a heatmap.

## RESULTS

### Demographic information of the TNBC patients

Our study includes 28 samples with adequate tissues together with complete clinical and histological information. These specimens were selected from our archival breast cancer specimens frozen in RNA*later*® solution based on negative immunohistochemical

**Table 1  Clinical and histopathological information of selected specimens.**

| Characteristics | Number of patients ($n$ (%)) |
| --- | --- |
| Average age at diagnosis (years ±SD) | 51.39 ± 12.86 |
| Histologic type | |
| –Invasive Ductal Carcinoma | 25 (89.29) |
| –Invasive Lobular Carcinoma | 2 (7.14) |
| –Others: Inflammatory Breast Cancer | 1 (3.57) |
| Histologic grade | |
| –Well-Differentiated | 0 (0) |
| –Moderately-Differentiated | 1 (3.57) |
| –Poorly-Differentiated | 27 (96.43) |
| Lymph node metastasis | |
| –Yes | 7 |
| –No | 4 |
| –Unknown | 17 |
| Ki67 (% ±SD) | 60.44 ± 16.8 |
| Lymphovascular Invasion ($n$ (%)) | |
| –Yes | 11 (39.29) |
| –No | 14 (50) |
| –Unknown | 3 (10.71) |

staining for ER, PR, and HER2, according to pathology reports. For samples with equivocal HER2 stain, only those with negative fluorescence *in situ* hybridization (FISH) results were included in our study. Our patient cohort's average age at diagnosis was 51.39 ± 12.86 years old. Most samples were invasive-ductal carcinoma, with poorly differentiated cancer with an average Ki67 staining of 60.44%. Demographic information of these specimens is shown in Table 1.

## Mapping Thai TNBC subgroups with previously established TNBC classifiers

We used the NanoString Breast 360 gene panel to profile gene expressions of samples from our cohort. This assay includes genes related to breast cancer molecular subtypes, TNBC signature, immunophenotypes, and DNA-repair functions. Results from unsupervised (k-means) clustering with the elbow method showed that the optimum number of groups was $k = 4$, which created the sharpest elbow and could explain up to 79.22% variance. We also determined silhouette scores and created silhouette coefficient plots. The results demonstrated that the silhouette scores at $k = 4$ and $k = 5$ were equal (0.45). However, the coefficients of the plot at $k = 4$ were all positive, while the coefficients of two samples at $k = 5$ were less than 0, indicating the possible incorrectly assigned samples in the clusters (Fig. 1A and see more in Fig. S1). As a result, we selected $k = 4$ to classify our 28 TNBC samples into four major subgroups, which we refer to from this point onwards as the SiSPTNBC Group1-4 (Figs. 1A–1B). We then attempted to classify our specimen subgroups to understand the similarity to the TNBC classification defined by *Lehmann et al. (2021)*,

which we refer to from this point onwards as Lehmann's subtypes. The matching result is presented as a Subclass Association (SA) matrix with a significant $p$-value for each SA, as shown in Fig. 1C. After performing normalization, batch analysis, and subclass mapping, we showed that our SiSPTNBC Groups 1–3 were highly correlated with LAR, BL2, and M of Lehmann's subtypes. Specifically, the SiSPTNBC Group 1 (seven samples (25%)) and Group 3 (10 samples (35.7%)) showed significant subclass matching with the LAR ($p = 0.02$) and M ($p = 0.04$) subtypes, respectively. Group 2 (six samples (21.4%)) aligned well with the BL2 ($p = 0.02$) subtype. Interestingly, the SiSPTNBC Group 4 (five samples (17.9%)) exhibited unique and uncorrelated expression patterns with any Lehmann's subtypes. The correlations with Lehmann's subtypes were confirmed, as illustrated in the heatmap (Fig. 1E). Samples with advanced stages (large tumor size and lymph node metastasis) were equally distributed among these four gene expression-based subgroups, showing no bias of patients' demographic in this study cohort (Fig. 1E). This finding thus emphasized that clinical information alone may not be adequate for the stratification of TNBC patients. Molecular data from the tumors provided more in-depth information about the disease for each patient, hence, better for further developing prognostic or predictive biomarkers in TNBC patients.

We also examined the similarity of our SiSPTNBC subgroups to the PAM50 molecular subtypes. The reference profiles were established from published data with known molecular subtypes (*Bustamante Eduardo et al., 2019*), together with gene expression data from our archival specimens (SiSP-ER+ samples and SiSP-HER2+ samples). Most SiSPTNBC samples were classified as a basal-like molecular subtype (Figs. 1D–1E). Other molecular subtypes (HER2-enriched, Luminal B, and Normal-like) were also detected in the TNBC samples. The molecular subtypes of SiSP-HER2+ (HER2-enriched) and SiSP-ER+ (Luminal A and B) samples matched their respective IHC subtypes, confirming that our normalization was accurate.

Interestingly, the second-most PAM50 subtype in our TNBC samples, the HER2-enriched subtype, came from the SiSPTNBC Group 1. Group 1 also showed similarity to the LAR Lehmann's subtype. Specimens with Luminal B and normal-like subtypes were found in SiSPTNBC Group 1 and Group 2, respectively. Our results agreed with the prior findings, showing that most TNBC was classified as the basal-like PAM50 molecular subtype. Additionally, the high number of HER2-enriched specimens within Group 1 was consistent with prior reports showing that most samples with HER2-enriched and Luminal subtypes also fell within the LAR TNBC subtype (*Ahn et al., 2016*; *Lehmann et al., 2021*).

## Differential cancer hallmark activation among different TNBC subgroups

The molecular uniqueness of different SiSPTNBC subgroups was examined following the MSigDB Hallmark Pathway database (*Liberzon et al., 2015*). The top-10 signature hallmarks of each SiSPTNBC group are shown in Fig. 2. We realized that some of the signature hallmarks, especially those in Group 1 and Group 3, such as estrogen response and EMT, seemed to be at lower significance. This could be due to the low number of samples and low RNA counts of some genes in the hallmarks. Nonetheless, we referred

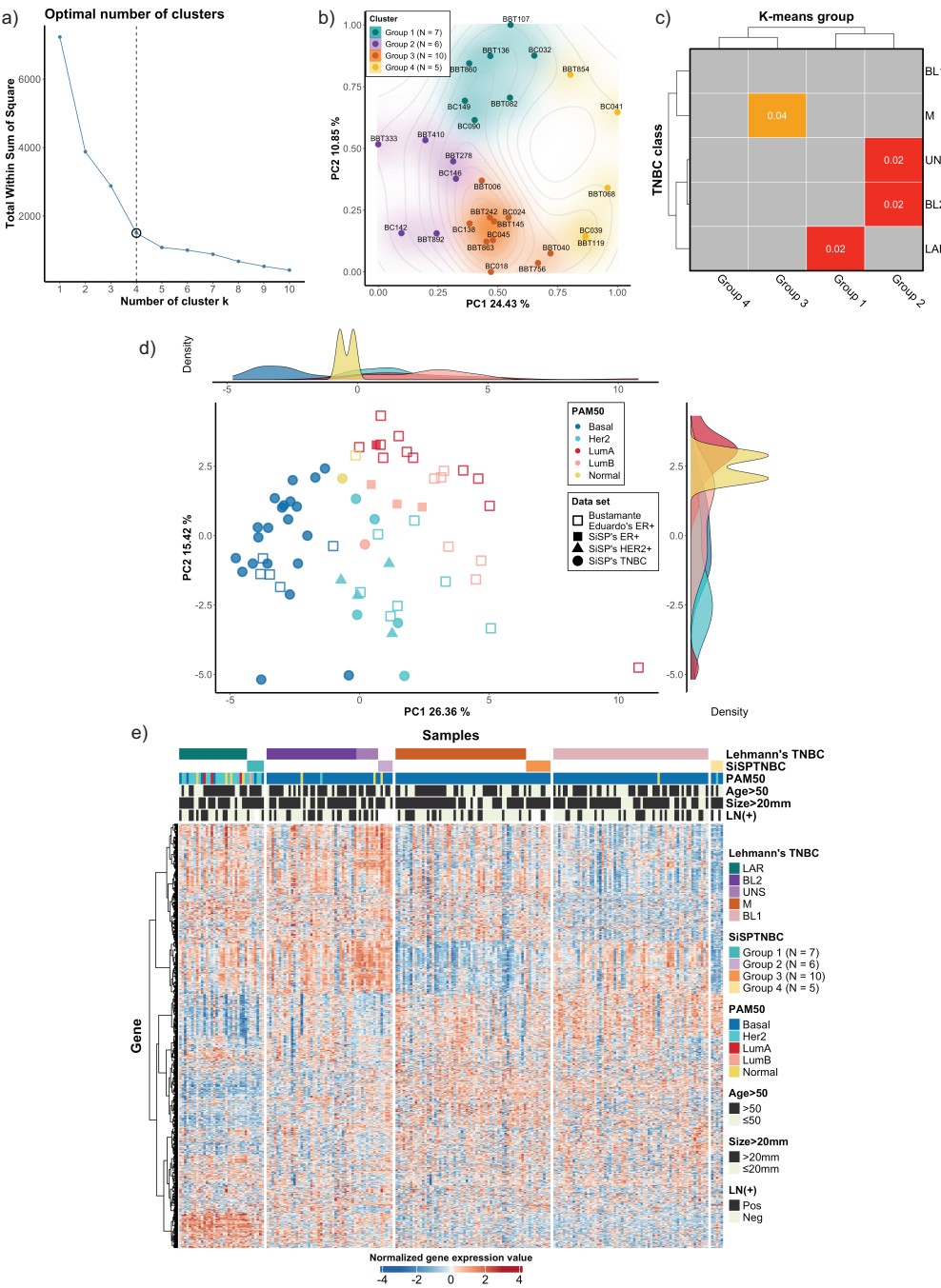

**Figure 1  Classification of TNBC subgroup (SiSPTNBC Groups 1-4) using gene expression data from NanoString nCounter® Breast Cancer 360 panel.** (A) An elbow plot was analyzed for selecting the optimal number of clusters for the k-means algorithm. K-means with cluster number equal 4, which gave the sharpest elbow with the stabilized total within the sum of squares after $k = 4$, was selected. (B) Principal Component Analysis (PCA) plot of the SiSPTNBC samples at $k = 4$ was annotated as clustered subgroups projected onto their first two principal components. (continued on next page...)

**Figure 1 (…continued)**
(C) Subclass Mapping (SubMap) analysis between SiSPTNBC groups and Lehmann's TNBC subtypes. Significant correlations were shown between SiSPTNBC Groups 1, 2, and 3 and Lehmann's LAR, BL2, and M subtypes, respectively. (D) PCA according to PAM50 subtypes of our breast tissue samples (filled symbols) and published data (unfilled symbols) from *Bustamante Eduardo et al. (2019)* was analyzed. The ER+, HER2+, and TNBC samples are annotated by square, triangle, and circle symbols, respectively. The majority of TNBC samples were basal-like PAM50 subtype, and the second-most frequent PAM50 subtype among TNBC samples was HER2-enriched. e) Heatmap of gene expression data from SiSPTNBC and TCGA samples clustered by Lehmann's subtypes SiSPTNBC subgroups was analyzed with PAM50 subtypes, age at diagnosis, gross tumor sizes, and lymph node status annotations (PC, principal component; LN, lymph node; Pos, positive; Neg, negative).

to these hallmarks since they are consistently found in Lehmann's subtypes. Regardless of the low significance, we only observed those hallmarks in certain groups. The signature hallmarks for SiSPTNBC Group 1 include cell proliferation and differentiation-related pathways (MYC targets, NOTCH signaling, WNT-$\beta$ catenin), metabolic-related pathways (cholesterol homeostasis, pancreas $\beta$ cell), and estrogen response pathway (Fig. 2). The enrichment in the hormone response and metabolic pathways agreed well with the molecular signatures of Lehmann's LAR subtype. Similar to previous publications, we found higher expression of *AR* (3.35-fold), encoding the androgen receptor and a vital hallmark gene of LAR subtype, in SiSPTNBC Group 1 compared with other groups (Fig. S3). Overall, the molecular characteristics of the Group 1 demonstrated the upregulation of hormone signaling and metabolic-related pathways.

SiSPTNBC Group 2 showed upregulated hallmark genes representing proliferation and cell cycle-associated pathways (PI3K/AKT/mTOR signaling, mitotic spindle, MYC targets, spermatogenesis, E2F targets, and G2M checkpoint), as well as apoptosis (Fig. 2). Some of these signatures (PI3K/AKT/MTOR signaling) were concordant with those found in Lehmann's BL2 subtype. Our results did not show the EMT signature typically found in the BL2 subtype. Interestingly, the IL-6/JAK/STAT3 and complement pathways were the most highly upregulated genes in Group 2 (Fig. S3), inferring an association with immune activation and cancer metastasis control, which is not reported in the BL2 subtype. The upregulation of immune-related pathways may thus infer its higher immune infiltration than other groups.

The critical hallmarks for SiSPTNBC Group 3 were stemness-related pathways (Hedgehog signaling, EMT), survival response such as ER stress response and detoxification (unfolded protein response, xenobiotic metabolism, peroxisome), as well as metabolic-related pathways (heme metabolism, fatty acid metabolism, bile acid metabolism) (Fig. 2). Enrichment of the stemness-related pathways is consistent with Lehmann's M subtype (*Aburjania et al., 2018*; *Lehmann et al., 2021*; *Lizarraga et al., 2016*; *Yin et al., 2020*; *Zhang et al., 2018*). However, the other hallmarks were not previously reported in the M subtype. Hence, in addition to the pathways related to stemness regulation, Group 3 also showed significant molecular signatures of metabolic and survival response pathways.

The most important molecular signature for SiSPTNBC Group 4 was mainly associated with immune-regulation pathways, especially those involved with immune cell recruitment and inflammatory responses (complement, IL2-STAT5 signaling, TNF-$\alpha$ *via* NF-$\kappa$B

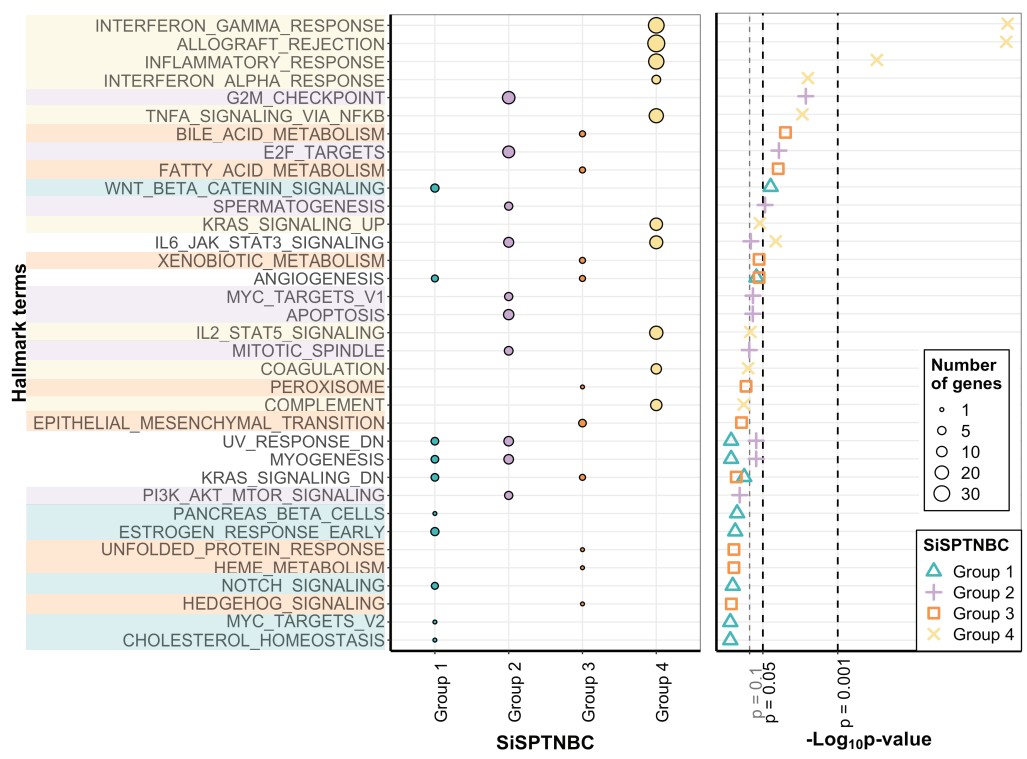

**Figure 2** **Gene set enrichment analysis of SiSPTNBC Groups 1–4.** The left panel showed the top 10 hallmarks of each SiSPTNBC subgroup ranked by *p*-values. The size of the circle is proportional to the number of genes in each pathway. The right panel showed levels of *p*-values ascendingly reordered. Three thresholds indicated *p*-values at 0.1, 0.05 and 0.001.

signaling, IFN-$\alpha$ response, inflammatory response, allograft rejection, IFN-$\gamma$ response). Other molecular hallmark signatures observed in Group 4 include the cancer progression-related pathways (coagulation and KRAS signaling) (Fig. 2). We also observed an increase in the gene expression of ECM proteins and E2F targets (Fig. S3), which are usually involved in cell migration and cell cycle control. Overall, SiSPTNBC Group 4 exhibited molecular characteristics of immune regulation and cancer progression-related pathways. We did not find a correlation between Group 4 and any of Lehmann's subtypes, and the signatures found in this group did not show similarities to any of Lehmann's subtypes.

## Comparison of immune cell composition and immunoblockade-related gene activation across different TNBC subgroups

Immune checkpoint inhibitors have been increasingly used in breast cancer patients, especially the TNBC subtype. Typical predictive biomarkers for immune checkpoint inhibitors (ICIs) include the IHC staining of immune checkpoint receptors such as PD-L1 and MMR IHC or NGS-based scoring of tumor mutational burden. Since the nCounter Breast 360 panel includes immune-associated genes, we were interested in comparing the difference of these immunomodulatory genes among the different SiSPTNBC groups

to examine their potential differences in immune cell infiltration and the activation of immunoblockade-related genes.

Group 1 specimens showed fewer cancer-associated fibroblasts (CAFs), endothelial cells, and neutrophils than Group 2 and Group 3 (Fig. 3; top). Interestingly, Group 1 tissues with HER2-enriched PAM50 subtype (BC149, BC032, BBT860) showed a lower amount of CD8+ T cells, myeloid dendritic cells, B cells, and cytotoxic lymphocytes than those with basal-like subtype (BC090, BBT136) (average ±SD arbitrary MCPCounter unit of HER2-enriched *vs.* basal-like samples: 40 ±21.46 *vs.* 166 ±92.24 for CD8+ T cells; 16 ±10.35 *vs.* 134 ±59.53 for B cells; 8.06 ±5.34 *vs.* 48 ±13.76 for myeloid dendritic cells; 57.28 ±25.24 *vs.* 392.14 ±213.2 for cytotoxic lymphocytes). Group 1 showed down-regulation of most immunomodulatory genes, except for the *PDCD1* (PD1-encoding gene), which exhibited upregulation in most samples. *CD274* (PD-L1-encoding gene) overexpressed in two basal-like tumors compared with the rest of the group (Fig. 3 and Fig. S4).

SiSPTNBC Group 3 showed the highest consistent signature with the M subtype, known for its low immune cell infiltration. However, we found that almost all tumors in Group 3 showed an increase in CAFs, endothelial cells, and neutrophils. Half of the tumors in this subgroup showed a high number of myeloid dendritic cells and CD8+ T cells, although not as high as Group 2. Tissues that were predicted to have a higher number of CD8+ T cells and myeloid dendritic cells consistently showed higher expression of the inflammatory cytokine and chemokine genes (*IL-6*, *CXCL8* (*IL-8*), *CCL2*, *CCL7*) with significant expressions in *CCL2* ($p = 0.00096$) and IL-6 ($p = 0.0074$) (Fig. S4) than those with lower T cells and dendritic cells. *CCL2* and *CCL7* encode monocyte chemotactic proteins 1 and 3, respectively, known to be strong recruiters of myeloid-derived cells, including monocytes, macrophages, dendritic cells, myeloid-derived suppressor cells, and lymphocytes, regulating tumor microenvironment. (*Jin et al., 2021*; *Liu et al., 2018*; *Zhang et al., 2020*). *IL-6* and *CXCL8*, important autocrines in TNBC, were known to activate neutrophils and NK cells migration to the tumor (*Baggiolini & Clark-Lewis, 1992*; *Wu et al., 2019*). The upregulation of *CCL2* and *CCL7* implies the high number of immune cells in the tumor. However, not all the recruited immune cells are cytotoxic cells. Studies reported that *CCL2* and *CCL7* upregulation also increased infiltration of suppressive immune cells, *e.g.*, MDSC, and caused a blockade of cytotoxic cell functioning (*Takacs et al., 2022*).

Interestingly, we observed anti-correlated immunophenotypic characteristics between Group 2 and Group 4. Group 2 showed a higher abundance of TME cells than the other groups. The higher immune cells in the Group 2 were concordant with its enrichment in immune-associated gene expression hallmarks. We also found higher cytotoxic lymphocytes in the samples within Group 2. On the contrary, Group 4 showed the lowest abundance of TME cells (Fig. 3; top). Group 4 also obtained fewer endothelial cells and CAFs. When examining the expression levels of genes associated with immune blockade (Fig. 3; bottom and Fig. S4), we found a significant increase in leukocyte and endothelial surface molecule-encoding gene *ICAM1* ($p = 0.03$), immune cell recruitment and regulation-related genes *CCL2* ($p = 0.00096$), *HGF* ($p = 0.00032$) and inflammatory cytokine gene *IL-6* ($p = 0.0074$) in immune cells-abundant Group 2 samples. We also found increased expressions of immune blockade genes *PDCD1* (PD-1) ($p = 0.03$) and *CD274* (PD-L1)

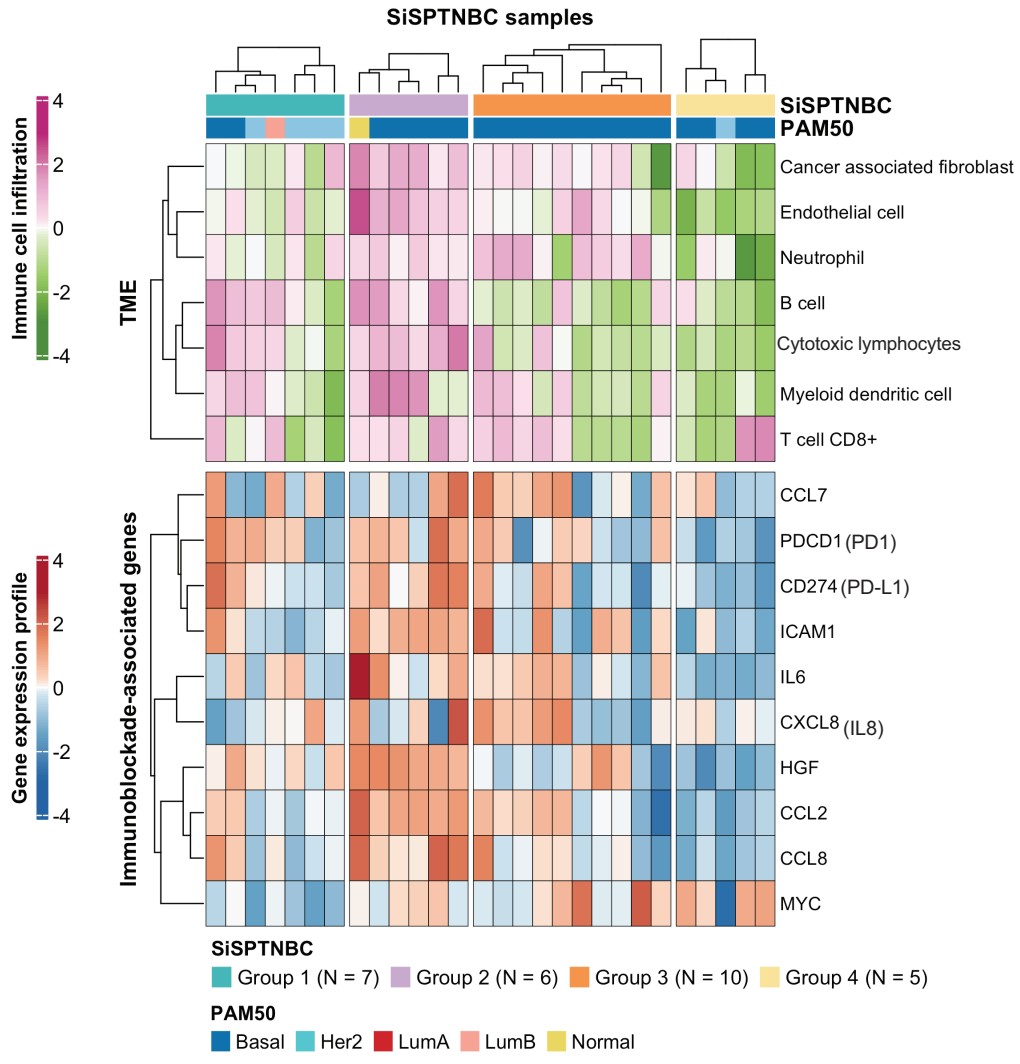

**Figure 3 Tumor microenvironment cell abundance and immune-associated genes in SiSPTNBC Groups 1–4.** A hierarchical clustering heatmap of the immune and stromal cell proportions from the MCPcounter algorithm and an expression profile of genes involved in an immunoblockade process is shown in the top and bottom panels, respectively. Annotations consist of SiSPTNBC subgroups and PAM50 prediction results (TME, tumor microenvironment).

($p = 0.013$) in Group 2. In the low immune cells-abundant Group 4, we observed consistent down-regulation of almost all the leukocyte membrane molecule, immune cell recruitment, and inflammatory cytokine genes (*ICAM1*, *CCL2*, *CCL8*, and *IL-6*) as well as the immune blockade genes *PDCD1* and *CD274*. However, most samples in Group 4 had higher *MYC* expressions than those in other groups ($p = 0.042$). This observation is consistent with the negative correlation between *MYC* gene expression levels and overall immune infiltration (*Lee et al., 2022*). However, due to our limited sample size, the benefits of ICIs to Group 2 patients still require further clinical validation.

## Comparison of DNA damage repair signature among different TNBC subgroups

Tumors with defects in HRR mechanisms are more sensitive to DNA-damaging drugs such as platinum-based chemotherapy and PARP inhibitors. Studies also showed that patients with impaired DNA double-strand break (DSB) repair pathways other than HRR might benefit from PARP inhibitors due to the synthetic lethality effect (*Liao et al., 2021*; *Wang et al., 2019*). It has been shown that over 21% of Thai TNBC patients carry the BRCAness genomic signature (*Niyomnaitham et al., 2019*). Thus, we were interested in investigating from gene expression profiles whether patients in our cohort exhibit molecular signatures of DNA damage repair defects that may infer sensitivity to PARP inhibition.

Specifically, we examined the expression of genes that function in homologous recombination repair (HRR), Fanconi anemia (interstrand cross-link repair), and non-homologous end-joining (NHEJ) repair pathways. Genes involved in the DNA replication and cell cycle controls, and other pathways that also influence DNA damage repair response were also included since prior studies reported their strong association with HRR deficiency (HRD) (*Friedberg et al., 2006*; *Lange, Takata & Wood, 2011*; *Liao et al., 2021*; *Peng et al., 2014*; *Wood et al., 2001*; *Zhuang et al., 2021*) (Fig. 4A). Group 1 showed reduced expressions of most genes in DSB repair, cell cycle regulation, and DNA replication pathways compared with other groups. The upregulated genes in Group 1 were involved with increased metabolism and survival of breast cancer cells (*Dobie & Skropeta, 2021*; *Inazu, 2014*; *Kohnz et al., 2016*), consistent with our previous cancer hallmark analysis. Tumors in Group 2 showed upregulations of some genes in HRR pathways (*i.e.*, *ATM*, *MUS81*, and *PTEN*). Tumors in Group 3 and Group 4 showed upregulation of the DNA repair mechanisms and cell cycle control genes across all samples (Fig. 4A), implying intact DSB repair processes of samples in these groups (*Peng et al., 2014*). When comparing the expressions of genes involved in the DSB pathways among the four subgroups (Kruskal–Wallis Test for global analysis and one-sided Wilcoxon rank-sum test for two-independent sample analysis) (Figs. 4B–4D), we observed dramatic differences at the single-gene level between some subgroups. The expressions of important genes in the HRR pathway, *ATM*, *MUS81*, and *PTEN*, were significantly higher in Group 2 ($p = 0.0071$, $p = 0.0043$, and $p = 0.016$, respectively). Two genes in the HRR pathway, *RAD51*, and *XRCC2*, cell cycle regulation (*CHEK2*), and Fanconi Anemia (*UBE2T*) were significantly upregulated in Groups 3 and 4 ($p = 0.042$, $p = 0.0042$, $p = 0.021$, and $p = 0.023$, respectively). Collectively, expressions of genes involved with HRR, Fanconi anemia, and NHEJ pathways were remarkably higher in Groups 2, Group 3, and Group 4. Group 1 showed consistently reduced expression of DNA damage repair and cell cycle regulatory genes.

Studies have shown that breast tumors with genomic HRD may show expression changes in DNA repair genes. Specifically, downregulation of *KIF2C*, *POLQ*, *POLD1*, *CCNE1*, *ATM*, *RAD54L*, *BLM*, *NETO2*, *FZD9*, *CXCL5*, and *VIT* and upregulation of *ADM*, *BTG2*, *C5orf38*, *CDKN1C*, *CYP4F3*, *ST6GALNAC2*, and *SLC44A4* are the reported signatures of HRD breast cancer (*Liao et al., 2021*; *Peng et al., 2014*). While not definitive, the gene expression patterns in our Groups 1 and 2 showed similar down-regulation of *KIF2C*, *POLQ*, *CCNE1*, *RAD54L*, *BLM*, *NETO2*, *FZD9*, *CXCL5*, and *VIT* and upregulation

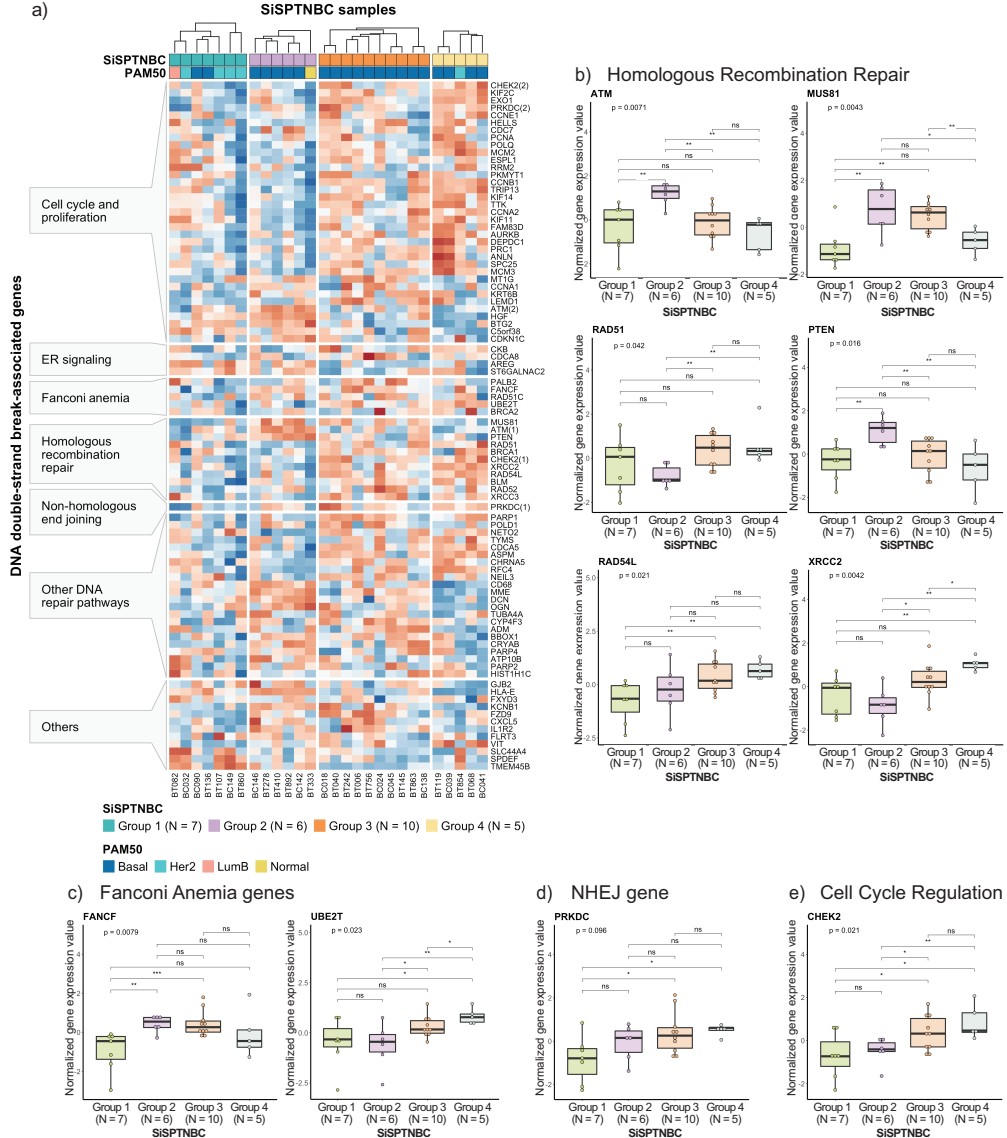

**Figure 4** **Analysis of gene expression profiles associated with DNA double-strand break repair processes in SiSPTNBC Groups 1–4.** (A) Heatmap demonstrates DSB repair-related gene expression from the SiSPTNBC samples. The genes on the $y$-axis were grouped according to their associated pathways. The SISPTNBC samples on the $x$-axis were grouped with annotated PAM50 subtypes. (B) Box plots compared the expressions of genes in a homologous recombination repair (HRR) pathway between the four SiSPTNBC groups. (C) Box plots compared the expressions of genes in Fanconi anemia pathway between the four SiSPTNBC groups. (D) Box plots compared the expression of a gene in a non-homologous end joining (NHEJ) pathway between the four SiSPTNBC groups. (E) Box plots compared the expressions of genes in cell cycle regulation between the four SiSPTNBC groups. The $p$-values of global and independent analyses were calculated by the Kruskal–Wallis test and the one-sided Wilcoxon Rank Sum test, respectively (not significant (ns): $p > 0.05$, $*$, $p \leq 0.05$, $**$, $p \leq 0.01$, and $***$, $p \leq 0.001$).

of *BTG2*, *C5orf38*, *CDKN1C*, *ST6GALNAC2*, and *SLC44A4*, implying that tumors from these two subgroups may exhibit HR loss-of-function. Nonetheless, genomic profiling and a larger sample size are required to confirm the benefit of PARP inhibitors for SiSPTNBC Group 1 and 2 patients.

## DISCUSSION

The aggressive TNBC is attributed to approximately 15% of breast cancer cases in Thailand, also diagnosed at a younger age than other populations (*Ding et al., 2019*; *Niyomnaitham et al., 2019*). Current treatments for all TNBC patients are similar, mainly relying on anthracycline- and taxane-based chemotherapy, despite known heterogeneity within the tumor and between patients. Even though some targeted therapies, such as ICIs and PARP inhibitors, have been approved for TNBC patients, reimbursement of such advanced regimens is still limited in Thailand, and many TNBC patients also responded poorly to such treatments (*Lehmann et al., 2011*; *Lehmann et al., 2021*; *Wang et al., 2019*).

In this study, we attempted to characterize the molecular subtypes of Thai TNBC patients based entirely on gene expression data from the NanoString nCounter® technology. Previous work that aimed to characterize the molecular subtypes of TNBC utilized other methodologies, including histology staining, gene-expression profiles from RNA sequencing or microarrays, and mutational analysis. The nCounter technology offers advantages over RNA sequencing and microarrays, such as the lack of reverse transcription and cDNA amplification steps, direct hybridization of mRNA to probes, and using a lower amount of RNA (*Narrandes & Xu, 2018*). We used a medium-size genes panel, Breast Cancer 360, to robustly classify subtypes on our TNBC cohort and gain insight into their signature hallmarks, immune abundance and immunoblockade-associated gene expressions, and signatures of DNA damage-related pathways, all done without DNA information. Despite the limitation on sample size, our study offers a practical approach to implementing gene expression-based information for the stratification of TNBC patients in the clinical setting.

Our study identified four subgroups from Thai TNBC patients, with only 3 subgroups showing similar profiles with Lehmann's subtypes. This finding is consistent with prior studies showing that TNBC patients from different ethnicities had particular molecular subtypes and were not necessarily the same as Lehmann's classification (*Ding et al., 2019*; *Jiang et al., 2019*). Since Lehmann's classification does not include immune-related subtypes (*Lehmann & Pietenpol, 2015*), we also compared our SiSPTNBC subgroups with the Burstein classification system (*Burstein et al., 2015*). SiSPTNBC Groups 2 and 3 showed correlated patterns with the MES and BLIS subtypes. Although Group 1 and Group 4 showed no similarity to other Burstein's subtypes (Fig. S2), we believe profiling the immune cell composition and the related immunomodulatory pathways is crucial for accurately predicting ICI benefits in TNBC patients.

Despite our small sample size, we demonstrated that investigating the underlying driver pathways could offer insights into specific treatment strategies among the different TNBC subgroups. For instance, patients in Group 1, like the LAR subtype, showed over 3-fold

upregulation of *AR*, implicating the possible sensitivity to AR inhibitors in this group. It has been shown that TNBC patients with LAR signature showed resistance to standard chemotherapy but sensitivity to enzalutamide (AR inhibitor) (*Yin et al., 2020*). Group 2 patients showed enrichment of genes involved in cell proliferation, similar to the BL2 subtype. Group 2 patients also showed enrichment of genes involved in cell cycle regulation and high immune cell abundance. Hence, Group 2 patients may have better responses to chemotherapy or ICIs. ICI treatment may not be recommended for patients in Group 4, where we observed low immune cell abundance.

Current treatment guidelines for ICI and PARP inhibitors rely on companion diagnostics that utilize standard IHC or DNA sequencing. For TNBC, the study by *Jiang et al. (2019)* compared HRD scores among Burstein's subtypes and PARP inhibitor response. BLIS subtype patients showed high HRD scores and sensitivity to PARP inhibitors with longer relapse-free survival intervals (*Burstein et al., 2015*; *Jiang et al., 2019*; *Marra et al., 2020*). Nonetheless, a broad comparison of PARP inhibitor response across all TNBC subgroups is still under clinical trials, such as the FUSCC Refractory TNBC Umbrella (FUTURE) study (NCT 03805399) (*Zhao et al., 2020*). Moreover, a prospective phenotypic-based study is needed to confirm the correlation between subtypes and responses to ICIs and PARP inhibitors.

Our investigation of TME cell composition and expression of immunoblockade-related genes implies that patients from different SiSPTNBC subgroups may exhibit a differential response to ICIs. Typically, TNBC patients contain higher mutations and more pronounced immune cell infiltration than other breast cancers (*Dieci, Miglietta & Guarneri, 2021*), and over 20% of TNBC patients showed upregulation of PD-L1 (*Oner et al., 2021*). Hence, using ICIs in combination with standard chemotherapy has been of interest to improve the clinical outcome for TNBC patients (*Kim, Choi & Lee, 2022*; *Qureshi et al., 2022*). Our study observed that Groups 1 and 4 and four showed fewer CAFs, endothelial cells, neutrophils, and CD8+ T cells. The increase in *MYC* expression in Group 4 was correlated with increased expression of genes involved in the cell cycle and reduced overall immune cell infiltration and antigen presentation. These implied that Group 4 patients might respond poorly to ICIs treatment. On the other hand, Group 2 showed more TME cells than other subgroups. This result concords with the upregulation of immune cell regulation genes *IL-6, CCL21,* and *CD36* in Group 2 samples. Specifically, high endothelial venule strongly predicted T and B cell infiltration (*Martinet et al., 2011*). Group 2 also exhibited increased expressions of immunoblockade-associated genes. Hence, the patients in Group 2 may benefit from the blockade of immune checkpoint receptors.

We also investigated expression signatures of genes involved in DSB repair pathways. Consistent expression patterns of HRD repair-related genes with HRD tumors from previous studies were observed in Groups 1 and 2, suggesting DNA repair deficiency. However, HRD status confirmation requires genomic HRD signatures, *i.e.,* HRD scores or Signature 3, which we could not calculate due to the lack of whole genome sequencing data. Our study focuses primarily on gene expression using nCounter® technology to classify TNBC. Additional studies, including WGS and PARP inhibitor response in patient-derived

models, will therefore be necessary to confirm DSB repair impairment and the benefits of PARP inhibitors to patients in each group.

The small sample size and inferences from bulk gene expression undoubtedly limit our study. While immune abundance analysis was predicted from gene expression using the MCPCounter algorithm, it still lacked the validation of cell location in the tumor microenvironment, which was shown as an essential marker for predicting treatment outcome (*Tsujikawa et al., 2020*). Nonetheless, the multiplex immune-staining process is more time-consuming and requires personnel with more advanced expertise than gene expression profiling. More in-depth analysis of gene expression profiles across different TME locations will be needed to characterize the heterogeneity of the hallmark signatures and immune cell infiltration in the other tumor locations.

## CONCLUSION

We used nCounter-based gene expression data to classify 28 Thai TNBC samples into four main subgroups (SiSPTNBC Group1-4) and to identify biomarkers predictive of innovative treatment regimens such as ICIs and PARP inhibitors. We showed a significant correlation of SiSPTNBC Groups 1, 2, and 3 to LAR, BL2, and M subtypes by Lehmann's TNBCtype-4 classification, respectively. Our results implicated that SiSPTNBC Group 1 patients may benefit from AR and PARP inhibitors, and those in Group 2 could benefit from ICIs. Our study provides preliminary evidence in utilizing nCounter-based gene expression profiling for stratifying TNBC patients and selecting appropriate treatment regimens in the clinical setting. Further clinical studies with larger cohort are needed to confirm the efficacy of PARP inhibitors and ICIs in TNBC patients of different subgroups.

## ACKNOWLEDGEMENTS

We would like to thank Prof. Dr. Pornchai O-charoenrat, Ms. Surat Phumphuang, and Ms. Pawinee Sukcharoen from the Division of Head Neck and Breast Surgery, Department of Surgery, Faculty of Medicine Siriraj Hospital for coordinating specimen collection and clinical data retrieval.

### Funding

This research project was supported by Mahidol University and The Foundation for Cancer Care Siriraj Hospital. Natini Jinawath is a recipient of mid-career research grant (# N41A640161) from the National Research Council of Thailand. The funders had no role in study design, data collection and analysis, decision to publish, or preparation of the manuscript.

### Grant Disclosures

The following grant information was disclosed by the authors:
Mahidol University and The Foundation for Cancer Care Siriraj Hospital.
The National Research Council of Thailand: #N41A640161.

## Competing Interests

The authors declare there are no competing interests.

## Author Contributions

- Monthira Suntiparpluacha conceived and designed the experiments, performed the experiments, prepared figures and/or tables, authored or reviewed drafts of the article, and approved the final draft.
- Jantappapa Chanthercrob analyzed the data, prepared figures and/or tables, authored or reviewed drafts of the article, and approved the final draft.
- Doonyapat Sa-nguanraksa performed the experiments, prepared figures and/or tables, and approved the final draft.
- Juthamas Sitthikornpaiboon performed the experiments, prepared figures and/or tables, and approved the final draft.
- Amphun Chaiboonchoe analyzed the data, prepared figures and/or tables, and approved the final draft.
- Patipark Kueanjinda analyzed the data, prepared figures and/or tables, and approved the final draft.
- Natini Jinawath conceived and designed the experiments, authored or reviewed drafts of the article, and approved the final draft.
- Somponnat Sampattavanich conceived and designed the experiments, authored or reviewed drafts of the article, and approved the final draft.

## Human Ethics

The following information was supplied relating to ethical approvals (i.e., approving body and any reference numbers):

The Siriraj Institutional Review Board approved this study (COA number Si 329/2017).

## Data Availability

The raw data is available in the Supplemental Files.

## Supplemental Information

Supplemental information for this article can be found online at http://dx.doi.org/10.7717/peerj.15350#supplemental-information.

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
