# Peer review of "Retrospective study of transcriptomic profiling identifies Thai triple-negative breast cancer patients who may benefit from immune checkpoint and PARP inhibitors"

_PeerJ, doi:10.7717/peerj.15350_

## Round 0.1 · original submission · Minor Revisions

This is a well-structured manuscript. The authors are expected to address all of the issues raised by the reviewer.

Reviewer 1 ·

Basic reporting

This is a well-written and clear manuscript examining gene expression-based profiling using Nanostring nCounter in 28 Thai TNBC patients. The results suggest four major TNBC subgroups and the potential use of immune checkpoint and PARP inhibitors in a group of TNBC patients.
While there is an inevitable overlap with existing classification signatures, the authors also report patients with unique and uncorrelated expression patterns. The more exciting part of the study is the immune checkpoint-related gene signatures.
The authors looked into the immune cell composition of the TNBC subgroups and described differences among patients. I have some minor issues that need to be clarified to improve the clarity of the work.

Experimental design

Research question and methods are well defined.

Validity of the findings

please see the comments below.

Additional comments

1. It would be nice to see the expression levels of some biomarker genes in patient samples (e.g., PDL1, MYC), to judge the mean expression values and how diverse the expression levels are in each group (a supplementary figure would be ok too)
2. Can you please expand the figure legends so that they are more self-explanatory?
3. Please also indicate patient numbers on bar graphs for patient groups (e.g. Figure 4)
4. A brief discussion on the method used (RNA-seq vs Nanostring ) would be informative to the readers
5. For expression data/bars, showing the actual data points rather than boxes would be more informative.

Reviewer 2 ·

Basic reporting

This manuscript was a research paper trying to classify Thai triple-negative breast cancer (TNBC) retrospective cohort subgroups. The authors used n-Counter-based Breast 360 gene expression to classify the cases. The cohort was classified into four major subgroups and the authors discussed the unique characteristic of each subgroup in the manuscript. The authors also pointed out the potential treatment solution for according to the subgroup characteristic. Although the study is limited by the small sample size and the analysis relies mainly on bulk gene expression, such data can serve as a good resource for further TNBC study.

The language of the manuscript is professional, clear and unambiguous.

Sufficient background information has been provided in the introduction.

Line 536 the DOI link format is different from others.

Experimental design

The experiment is well-designed. The grouping is mostly based on the transcriptomic background of the TNBC samples. It would be better if the author include more clinical information or relevance to the four subgroups, it would be better. In addition, if the author can include more explanation of the mechanism, it would be more convincing why the certain group would benefit from immune checkpoint inhibitors and PARP inhibitors.

Validity of the findings

The dataset serves as a very good resource for further TNBC research. The conclusions are well stated, linked to the research question and limited to supporting results.

Reviewer 3 ·

Basic reporting

Language used is generally appropriate, but the article can use some minor polishing for typos and grammar.

Experimental design

Methodology and data analysis generally appropriate and up-to-date.

Validity of the findings

Conclusions may be slightly overstated given limitations in sample size. Conclusions regarding associations with Lehmann's subtypes should also be toned down as the data is inconsistent.

Additional comments

In this study, the authors conduct a gene expression analysis of a fairly small number of TNBC samples from the Thai population using NanoString nCounter technology. The authors use the gene expression data to classify their samples into four subtypes, and proceed to compare their classification with other classification systems from literature. The authors continue by conducting analyses of biological pathways, immune cell infiltrates, and DNA repair genes between their four subtypes, with heavy reference and comparisons to Lehmann’s subtypes from literature. Overall, I find that their methodology and conclusions drawn to be mostly appropriate for the scope of their study. The authors should also be commended for taking the time and effort to do an in-depth data analysis that incorporates significant batch correction and normalization efforts as well as comparisons with external datasets. The figures are generally well-constructed as well. However, given the small sample size and some discordance between the results presented, I believe that the results of this study should be treated as preliminary rather than conclusive, and thus presented with more caution.

Main comments:
1. The small sample size is a significant limitation when it comes to drawing conclusions regarding the number and prevalence of subtypes, and even more so when trying to draw conclusions regarding differences between these subgroups. The authors do mention this briefly as a limitation, but I believe more caution should be applied to concluding statements throughout the article.
2. The associations between the author’s own subgroups with Lehmann’s subtypes are not very convincing. They appear to be drawn largely from a single analysis of subgroup mapping, whereas other analyses such as prevalence estimates, pathway analyses and immune cell infiltration are inconsistent or even contradictory. Group 1 is plausibly associated with LAR, but the other reported associations seem to me to be quite tenuous. The authors should perhaps focus more on describing their own results rather than trying to fit their results into Lehmann’s categories. Also, it might be beneficial to consider Burstein’s subtypes as a better fit throughout given that Burstein has immune-high and immune-low subtypes similar to the authors’ subgroups.
3. The suggestion that Group 1 may better respond to PARP inhibitors should be made with more caution as lower expression of DNA repair genes does necessarily indicate loss of function of HR pathways.
4. The right panel of Figure 2 is difficult to interpret as it seems that a lot of the pathways have marginal p-values – the authors should indicate where the threshold is for p=0.05 and p=0.001 on the graph.
5. “This finding emphasized that the clinical patterns alone cannot be used for predicting disease progression or severity, and molecular data from the tumors would provide more in-depth information about the individual patient’s disease, hence, a more robust predictor of prognosis.” This statement is inappropriate without any comparison of clinical features and molecular features to prognosis/progression/severity, which was not part of the study as far as I can tell.

Minor comments:
1. The UNS subtype is not a real subtype as it refers to samples with unstable classification, and thus should generally be excluded from any conclusive statements.
2. A lot of the results section describes the methodology used to do the data analysis. Some of this repeats what is in the methods section, but some of the methodology seems to only be described in the results section. Thus, the methods sections should be expanded to include all the methodological steps taken, while the description of the methods in results section can be shortened and summarized better.
3. The article as a whole is generally well written up and understandable, but can use a bit of polish to fix typos and grammar throughout.
4. Line 155 – “Most samples were intra-ductal carcinoma” - I believe the term here should be invasive ductal carcinoma

Reviewer 4 ·

Basic reporting

Overall the manuscript is well-structred and clearly writen.

Experimental design

The design and the explanation of the methods are well defined.

Validity of the findings

As I mentioned in the "Additinal Comments" in detail the authors need to emphasize the novelty of the fndings more.

Additional comments

In this manuscript the authors classified the TNBC samples and charactarized each subgroup according to different molecular markers and compared these subgroups with the different classification methods in the literature. The manuscript has importance in the field since the TNBCs benefit less from the treatment strategies and molecular calssification of TNBCs may increase the benefit but the highligted points below have to be reconsidered;

The authors found out that they have 4 subgroups within their Thai TNBC tissue cohort using unsupervised (k-mean) clustering with an elbow method. In figure 1a they have to explain more why they chose 4 subgroups instead of 5 in the legend of the figure or in the results section.

Since the resolution of the figure 1d is not good enough it is not possible to interpret the figure properly. By reading the results section the findings became clear. The resolution of the figure has to be improved.

Overall, the manuscript is lacking the idea of what novelty it brings to the literature. The authors combined the data obtained from different point of views like, comparison of DNA damage repair signatures, immune cell composition and checkpoint receptor activation and differential cancer hallmark activation among defined TNBC subgroups by them, but the results are not cohesive. They have to emphasize the combined power of the data on classification of TNBC samples and the novelty compared to the other methods.

---

## Round 0.2 · accepted · Accept

Thank you for the careful revision. The manuscript can now go forward for publication.

The Section Editor noted:
> I think that Table 1 needs to be reformatted - I am not sure what is in the second column, and this is not a typical table with headings.

Reviewer 3 ·

Basic reporting

No further comment, I am happy with the changes to the manuscript.

Experimental design

No further comment, I am happy with the changes to the manuscript.

Validity of the findings

No further comment, I am happy with the changes to the manuscript.